# pH and Salt-Assisted Macroscopic Chirality Inversion of Gadolinium Coordination Polymer

**DOI:** 10.3390/molecules28010163

**Published:** 2022-12-25

**Authors:** Ting Hou, Lan-Qing Wu, Yan Xu, Song-Song Bao, Li-Min Zheng

**Affiliations:** 1State Key Laboratory of Coordination Chemistry, School of Chemistry and Chemical Engineering, Collaborative Innovation Center of Advanced Microstructures, Nanjing University, Nanjing 210023, China; 2Institute of Information Engineering, Suqian College, Suqian 223800, China

**Keywords:** coordination polymers, salt-assisted, chirality inversion

## Abstract

The precise adjustment of handedness of helical architectures is important to regulate their functions. Macroscopic chirality inversion has been achieved in organic supramolecular systems by pH, metal ions, solvents, chiral and non-chiral additives, temperature, and light, but rarely in coordination polymers (CPs). In particular, salt-assisted macroscopic chirality inversion has not been reported. In this work, we carried out a systematic investigation on the role of pH and salt in regulating the morphology of CPs based on Gd(NO_3_)_3_ and *R*-(1-phenylethylamino)methylphosphonic acid (*R*-pempH_2_). Without extra NO_3_^−^, the chirality inversion from the left-handed superhelix ***R*-M** to the right-handed superhelix ***R*-P** can be achieved by pH modulation from 3.2 to 3.8. The addition of NaNO_3_ (2.0 eq) at pH 3.8 results in an inversion of chiral sense from ***R*-P** to ***R*-M** as a pure phase. To our knowledge, this is the first example of salt-assisted macroscopic helical inversion in artificial systems.

## 1. Introduction

Helical architectures are ubiquitous in nature and display unique functions and topological properties [1,2,3,4]. Due to the uncertainty of chiral transcription and translation, the induction and inversion of helical structures is one of the most complex processes in biological systems [5]. Therefore, achieving multi-level assembly of helical structures from molecular to macroscopic levels in artificial systems with controllable macroscopic helical sense and elucidating the underline mechanism have become one of the hot spots in the research of chiral materials [6]. To date, the study of modulating the helical sense of the macroscopic helical structures via external stimuli has received increasing attention. Chiral inversion that can be achieved by pH [7,8,9,10,11,12], metal ions [13,14,15,16,17,18], solvents [19,20,21,22,23,24], chiral and non-chiral additives [25,26,27,28,29,30,31,32], temperature [33,34], light [35,36] have been reported, which are often associated with the modulation of non-covalent interactions, such as hydrogen bonds, van der Waals forces, and electrostatic interactions. However, it is worth noting that salt-mediated modulation of the chiral sense of helical structures has not been reported.

Salt plays an important role in living systems. Moderate salt intake can alleviate or prevent some diseases and improve body functions [37,38,39]. In addition, salt can alter non-covalent interactions, which may affect the self-assembly process. For example, high concentrations of salt can modulate the strength of electrostatic interactions between charged groups, destroy hydrogen bonds in protein secondary structure, cause changes of the α-helix [40,41] and β-sheet [42,43], or disrupt the structure of DNA, leading to structural changes from Z-DNA to Z’-DNA, or B-DNA to Z-DNA [44]. Various salts can also be used as additives to control the size, morphology, structure of inorganic crystals [45,46,47,48,49]. However, salt has rarely been used to control the morphology and structure of coordination polymers (CPs). The only example was associated with the Er/*R*, *S*-pempH_2_ [*R*, *S*-pempH_2_ = *S*-(1-phenylethylamino)methylphosphonic acid)] system, where the addition of salt led to a change in the crystal structure from a triple-stranded helix to a quadruple-stranded helix [50]. To our knowledge, salt effect on the morphology and helical sense of chiral CPs has not been documented. 

Our previous work has shown that the synergistic effect of pH and anion is crucial for the formation and helical sense of the macroscopic superhelices of the Tb(NO_3_)_3_/*R*, *S*-pempH_2_ and GdX_3_/*R*, *S*-pempH_2_ (X = Cl^−^, Br^−^, I^−^, CH_3_SO_3_^−^, C_6_H_5_SO_3_^−^, CF_3_SO_3_^−^) systems [51,52]. The pH can modulate the degree of deprotonation of the phosphonic acid moiety and, thus, the ratio of coexisting different types of chains, while the anion can modulate the interactions between chains and, thus, the formation and chiral sense of the superhelices. Considering the unique structure of the nitrate ion, a triangular-shaped singly charged oxygen anion, it not only can provide interactions through coordination bonds [53,54,55,56], but also act as a three-directional hydrogen receptor that affects the assembly of molecular building blocks [57,58,59,60]. Therefore, we hypothesize that the addition of extra nitrate salt with different concentrations into the Ln(NO_3_)_3_/*R*, *S*-pempH_2_ system may tune the interchain non-covalent interactions, disturb the self-assembling environment, and provide a driving force to control the chiral sense of resulting helical products. 

In this work, we chose Gd(NO_3_)_3_/*R*-pempH_2_ and systematically investigated the role of pH and salt in regulating the morphology of the products (Figure 1). We found that pure phases of blocky crystals of (H_3_O)[Gd_6_(*R*-pempH_2_)_3_(*R*-pempH)_15_](NO_3_)_4_·9H_2_O (***R*-Block**) were obtained at pH 2.6–2.9, and rod-like crystals of [Gd(*R*-pempH)_3_].1.5H_2_O (***R*-Rod**) at pH 4.2–4.6. When the pH was in between, left-handed superhelices (***R*-M**) appeared as a minor phase at pH 2.9–3.2, while right-handed superhelices (***R*-P**) were isolated as a pure phase at pH 3.6–4.2. Interestingly, the addition of 2.0 eq NaNO_3_ salt to the reaction mixture at pH = 3.8 leads to the chirality inversion of superhelices from right-handed to left-handed. This is the first example of chirality inversion of helical structures of coordination polymers by changing the concentration of salt. 

## 2. Results

### 2.1. pH Effect on the Reaction Product 

The hydrothermal reaction of a mixture of Gd(NO_3_)_3_·6H_2_O and *R*-pempH_2_ (molar ratio 1:5) at 120 °C for 24 h resulted in products with differet morphologies depending on the pH of the reaction mixture (Figure 2, Figure 1 and Appendix A). We obtained a pure phase of block-like crystals of (H_3_O)[Gd_6_(*R*-pempH_2_)_3_(*R*-pempH)_15_](NO_3_)_4_·9H_2_O (***R*-Block**) at pH 2.6–2.9. When the pH was 3.0–3.2, a mixture of left-handed superhelices (***R*-M**, minor phase) and block-like crystals (***R*-Block**) was observed. When the pH was raised to 3.7–4.2, a pure phase of right-handed superhelices (***R*-P**) was collected. At an even higher pH (4.3–4.6), rod-like crystals of [Gd(*R*-pempH)_3_]·2H_2_O (***R*-Rod**) were isolated. It is apparent that pH plays an important role in the self-assembly process and the chirality transcription and amplification of the system. This phenomenon has been previously observed in the Tb(NO_3_)_3_/*R*-pempH_2_ system [51]. However, the observation of left-handed superhelices of ***R*-M** as a minor phase at pH = 3.0–3.2 is unprecedented. To exclude the possible effect of metal ions, we conducted the same reaction but using Tb(NO_3_)_3_ to replace Gd(NO_3_)_3_. Left-handed superhelices were again obtained as a minor phase (Appendix A). Noting that the experimental condition such as the filling degree of the autoclave (ca. 53%) is not exactly the same as we used in previous work (>90%), we conducted similar hydrothermal reactions of Gd(NO_3_)_3_·6H_2_O and R-pempH_2_ but with different filling degrees (30, 50, 60, 70, 90%) at pH 3.2. As shown in Appendix A, the left-handed superhelices of ***R*-M** appeared together with ***R*-Block** when the filling degree was 30 and 50%, and vanished when the filling degree was 60%. When the filling degree was 70 and 90%, only right-handed superhelices of ***R*-P** were obtained. As the filling degree is directly related to the auto-generated pressure inside the autoclave, the above results indicate that pressure also plays an important role in the formation of the superhelices of Gd/*R*, *S*-pempH_2_ products. When the filling degree is ca. 50%, left-handed superhelices of ***R*-M** can form, and the chirality inversion can be achieved by adjusting the pH from 3.0–3.2 to 3.7–4.2. 

### 2.2. Structures of **R-Block** and **R-Rod**

Structural analyses showed that ***R*-Block** and ***R*-Rod** are isostructural to their Tb analogues [51]. ***R*-Block** crystallizes in the monoclinic system, with chiral space group *P*2_1_. The asymmetric unit contains three Gd^3+^, three *R*-pempH_2_, fifteen *R*-pempH^−^, four NO_3_^−^ anions, one H_3_O^+^ and eleven lattice water molecules (Appendix A and Appendix A). There are two kinds of positively charged chains in the structure. Chain I has the composition [Gd_3_(*R*-pempH_2_)_2_(*R*-pempH)_7_]^2+^, where three distinct Gd atoms, two different *R*-pempH_2_ and seven different *R*-pempH^−^ are found. Each Gd atom is eight-coordinated by oxygen atoms from the phosphonate groups. The Gd-O bond lengths lie in the range 2.310(5)–2.631(5) Å, and the O-Gd-O bond angles are 58.6(2)–157.4(2)° (Appendix A). The amino groups in the ligands are all protonated, while the phosphonate groups in the two R-pempH_2_ ligands are singly protonated at O18 and O27, but those in the seven *R*-pempH^−^ ligands are fully deprotonated. The Gd atoms are triply bridged by one O-P-O and two μ_3_-O(P) units, forming an infinite triple-stranded chain running along the b-axis (Figure 2). The pitch and diameter are 24.21 and 18.88 Å, respectively, for Chain I (Figure 2). The protonated phosphonate oxygen atoms O18 and O27 are involved in hydrogen bonding with the counteranion NO_3_^−^ [O27…O63: 2.545(8) Å, O18…O55: 2.462(19) Å] (Figure 3a). 

Chain II has the composition [Gd_3_(*R*-pempH_2_)( *R*-pempH)_7_]^+^ which contains one less *R*-pempH_2_ than Chain I. In addition, one of the three Gd atoms (Gd4) in Chain II is seven-coordinated and the other two (Gd5, Gd6) are eight-coordinated, unlike Chain I in which all Gd atoms are eight-coordinated. As a result, the Gd atoms in Chain II are bridged by one O-P-O and two μ_3_-O(P) units forming a trimer, and then the trimers are further linked by two O-P-O and one μ_3_-O(P) units forming a triple-stranded infinite chain (Figure 2). Again, the phosphonate group in *R*-pempH_2_ is singly protonated at O30 and is involved in hydrogen bonding with the nitrate anion. Compared to Chain I, the pitch of Chain II (24.21 Å) is the same, but the diameter (17.91 Å) is smaller (Figure 2). 

The positively charged Chains I and II are stacked in the lattice. Between the chains, there are four crystallographically different NO_3_^−^ anions and one H_3_O^+^ to balance the charge, as well as lattice water molecules. Therefore, an extensive hydrogen bonding network is found among the phosphonate oxygen atoms, the amino groups, the NO_3_^−^ anions, and the water molecules (Figure 3a). The shortest interchain distance is 15.334 Å (Appendix A). 

The structure of compound ***R*-Rod** was previously reported [52]. It crystallizes in the hexagonal systems, chiral group *P*6_5_, and has a general formula of [Gd(*R*-pempH)_3_]·1.5H_2_O. Only one distinct Gd atom is observed in the structure, which is eight-coordinated by oxygen atoms from six *R*-pempH^−^ (Figure 2). The Gd-O bond lengths lie in the range 2.285(13)–2.606(14) Å, and the O-Gd-O bond angles are 58.7(4)–161.1(4)°. ***R*-Rod** shows a neutral chain structure, where the Gd atoms are triply bridged by one O-P-O and two μ_3_-O(P) units. Compared to ***R*-Block**, although the pitch of the resulting triple-stranded chain (Chain III) in ***R*-Rod** is similar (24.11 Å vs. 24.21 Å in Chain I and II), the chain diameter is much shorter (16.56 Å vs. 18.88 Å in Chain I and 17.91 Å in Chain II) (Figure 2). The inter-chain distance is 15.756 Å. The lattice water molecules are located close to the chains and form hydrogen bonds with the phosphonate oxygen and amino nitrogen atoms [O1W…O6: 2.770(4) Å, O1W…O7: 2.970(3) Å, N3…O1W: 2.880(3) Å]. The interchain interactions are dominated by van der Waals contacts (Figure 3b). 

### 2.3. Characterization of Superhelices **R*-M*** and **R*-P***

Since superhelix ***R*-M** was obtained as a minor phase together with block-like crystals, we collected the sample by manual separation, while ***R*-P** was obtained as a pure phase at pH 3.8. To determine the composition of ***R*-M** and ***R*-P**, we performed energy dispersive X-ray spectroscopy (EDS), thermogravemetic (TG) ananlysis and elemental analysis. The EDS measurements revealed a molar ratio of Gd:P = 1:3 in both cases (Appendix A). By combining with the TG and elemental analysis results (Appendix A, Appendix A), we propose that the molecular formulae of ***R*-M** and ***R*-P** are Gd[(*R*-pempH)_2.70_(*R*-pempH_2_)_0.30_](NO_3_)_0.30_·2H_2_O and Gd[(*R*-pempH)_3_]·2H_2_O, respectively. 

Noting that the chemical composition of ***R*-P** is similar to that of ***R*-Rod** and that both products are obtained at very similar pH conditions, we envisage that the structure of ***R*-P** is closely related to that of ***R*-Rod**. Indeed, the PXRD patterns of both are very similar except that the peaks on the ***R*-P** pattern are all shifted to the left (Figure 4a). The result suggests that ***R*-P** has a similar structure to ***R*-Rod**, but with an enlarged unit cell volume. This is reasonable because the chain stacking in the superhelix ***R*-P** may be looser than that in the crystal ***R*-Rod**. To obtain the cell parameters of ***R*-P**, we indexed the diffraction peaks of ***R*-P** and ***R*-Rod** using TOPAS 5.0 [61], yielding the parameters: space group *P*6_5_, a = 16.34 Å, c = 24.40 Å, V = 5641.98 Å^3^ for ***R*-P**; and *P*6_5_, 15.86 Å, c = 24.23 Å, V = 5298.37 Å^3^ for ***R*-Rod** (Appendix A). Compared to ***R*-Rod**, the cell volume of ***R*-P** is expanded by ca. 343 Å^3^ (5641.98 Å^3^ vs. 5298.37 Å^3^ in ***R*-Rod**). 

Although the PXRD patterns of ***R*-P** and ***R*-Rod** are slightly different, their infred (IR) spectra are identical (Figure 4b). Both show a series of peaks between 900 and 1200 cm^−1^, assigned to the ν(P-O) stretching vibrations. The solid-state circular dichroism (CD) spectra are also identical, and both exhibit a negative Cotton effect at 262 nm (Figure 4c), attributed to the π-π* transitions of the chiral phosphonate ligand [62]. These results corroborate the structural similarity and chiral nature of ***R*-P** and ***R*-Rod** at the molecular level. Strong evidence to distinguish the difference between ***R*-P** and ***R*-Rod** is their vibrational circular dichroism (VCD) spectra. As shown in Figure 4d, ***R*-P** exhibits a strong negative VCD signal at 1014 cm^−1^ and three weak positive VCD signals near 1048, 1030, and 990 cm^−1^. In contrast, the peak at 1014 cm^−1^ is absent in the VCD spectrum of ***R*-Rod**. Clearly, the helical morphology of ***R*-P** is reflected by its VCD spectrum. 

Unlike ***R*-P**, the left-handed superhelix ***R*-M** was obtained at a lower pH and its formula contains 0.3 NO_3_^−^ anions per Gd. The IR spectrum of ***R*-M** is similar to that of ***R*-P** except for the enhanced intensity of the peak at 1385 cm^−1^, attributed to the stretching vibration of the un-coordinated NO_3_^−^ (Figure 4b). The results suggest that the phosphonate groups in ***R*-M** are partially protonated, forming positively charged chains which are balanced by nitrate anions. The intercalation of nitrate anions may increase the interchain distance and expand the cell volume. This is confirmed by the PXRD measurements. Compared to ***R*-P**, ***R*-M** shows an identical PXRD pattern, but the diffraction peaks are all left-shifted (Figure 4a). A Pawley fit of the PXRD pattern of ***R*-M** using TOPAS 5.0 led to the unit cell parameters: space group *P*6_5_, a = 16.73 Å, c = 40.30 Å, V = 9767.22 Å^3^ (Appendix A). The c-axis is significantly enlongated, which can be explained by the possible symmetry reduction of the chain structure due to the incorporation of nitrate anions. The CD spectrum of ***R*-M** is similar to that of ***R*-P** (Figure 4c), in agreement with the existence of the same chiral ligand in the two materials. Interestingly, the VCD profile of ***R*-M** is almost a mirror of that of ***R*-P**, where a positive signal was found near 1014 cm^−1^ and three weak negative signals near 1048, 1030 and 990 cm^−1^ (Figure 4d). This result is consistent with their macroscopic helix structure, i.e., the ***R*-P** superhelix is right-handed, while the ***R*-M** superhelix is left-handed. 

The formation of the ***R*-M** and ***R*-P** superhelices with similar chemical composition and structure but opposite handedness demonstrates that helical inversion may be achieved for the Gd(NO_3_)_3_/*R*-pempH_2_ system by adjusting the pH of the reaction mixture. When the pH is low, the phosphonate groups are partially protonated resulting in positively charged chains. The incorporation of a suitable amount of nitrate anions may affect the chain packing in the lattice and, thus, the formation of left-handed superhelix ***R*-M**. A slight increase in the pH leads to the deprotonation of the phosphonate groups and the decrease in the amount of nitrate anions. As a result, right-handed superhelix ***R*-P** is obtained in order to maximize the van der Waals contacts. The same phenomenon was previously observed for the GdX_3_/*R*-pempH_2_ (X = I, CF_3_SO_3_^−^) system [52]. Noting that ***R*-M** appeared as a minor phase at pH 3.2, we envisage whether it is possible to isolate a pure phase of ***R*-M** and accomplish chirality inversion through a combined effect of pH and salt. 

### 2.4. Salt Effect on the Chirality Inversion of Superhelices 

We first investigated the effect of additional NaNO_3_ on the product of the Gd(NO_3_)_3_/*R*-pempH_2_ reaction mixture. The above-mentioned results have shown that a pure phase of right-handed superhelix ***R*-P** can be obtained at pH 3.8 without the addition of extra NaNO_3_. By keeping the initial pH at 3.8, we conducted similar hydrothermal reactions of Gd(NO_3_)_3_/*R*-pempH_2_ at 120 °C for 24 h, but with the addition of different amounts of NaNO_3_. As shown in Figure 5 and Appendix A, right-handed superhelices dominated when the added NaNO_3_ was below 0.8 eq (vs. Gd(NO_3_)_3_). When the amount of NaNO_3_ was 1.2 eq, left-handed superhelices emerged together with nanofibers without clear helicity. When the amount of NaNO_3_ was further increased to 2.0–3.0 eq, a pure phase of left-handed superhelix (named as ***R*-M’**) was collected. When the amount of NaNO_3_ was above 4.0 eq, block-like crystals ***R*-Block** were obtained together with ***R*-M’**. It is clear that chirality inversion of the Gd(NO_3_)_3_/*R*-pempH_2_ system can be achieved by controlling the amount of extra NaNO_3_ salt at pH 3.8. 

Except for the morphology change, the salt-induced chirality inversion from ***R*-P** to ***R*-M’** was accompanied by other changes. The C:N molar ratio decreased from 8.9:1 to 7.8:1 when adding 0.0–2.0 eq NaNO_3_ (Appendix A), suggesting the incorporation of nitrate anions. This is confirmed by the IR spectra which show an increase in peak intensity at 1385 cm^−1^ (Appendix A). The PXRD patterns showed a slight left-shift of the peaks (Appendix A). Although the CD spectra remained similar (Appendix A), a significant change was observed in the VCD spectra (Appendix A). With 0.0 eq NaNO_3_, we obtained ***R*-P** which exhibits a strong negative VCD signal at ca. 1014 cm^−1^. This peak intensity was reduced for the product with 0.4 eq NaNO_3_. Interestingly, the sign of this signal became opposite with 0.8 eq NaNO_3_, and the intensity increased for the products with increasing amount of NaNO_3_. 

To examine whether ***R*-M’** is the same as ***R*-M**, we characterized the products by different techniques. Both the IR spectrum and PXRD pattern of ***R*-M’** are identical to those of ***R*-M** (Appendix A). The Pawley fit of the PXRD pattern of ***R*-M’** led to similar unit cell parameters: space group *P*6_5_, a = 16.82 Å, c = 40.37 Å, V = 9890.22 Å^3^ (Appendix A). The elemental and thermal analysis confirmed that **R-M’** has a chemical composition of Gd[(*R*-pempH)_2.70_(*R*-pempH_2_)_0.30_](NO_3_)_0.30_·2H_2_O (Appendix A and Appendix A), in agreement with that for ***R*-M**. In addition, both CD and VCD profiles of ***R*-M’** coincide with those of ***R*-M** (Appendix A). Based on these results, we conclude that ***R*-M’** is the same as ***R*-M**, both in composition and chiral structure. 

We next questioned whether the cation or the anion in the added NaNO_3_ salt plays a key role in the chirality inversion of the Gd(NO_3_)_3_/*R*-pempH_2_ reaction product. We conducted the same reactions at pH 3.8, but with adding different salts of Ba(NO_3_)_2_, KNO_3_ and NaNO_3_ (2.0 eq NO_3_^−^). In all cases, we obtained the same product ***R*-M** (Appendix A), indicating that the cation of salt does not affect the final product. We then conducted the same reactions at pH 3.8, but with the addition of 2.0 eq salts of NaCl, NaBr, NaNO_2_, and Na_2_SO_4_. We obtained a mixture of right-handed superhelix ***R*-P** and non-twisted fibers for NaCl and NaBr. For NaNO_2_ and Na_2_SO_4_, we obtained unrecognized new phases with of nanoplate and nanoparticle morphologies, respectively (Appendix A). The results corroborate that it is the nitrate anion of the salt that plays a vital role in directing the chirality inversion from ***R*-P** to ***R*-M**. 

To further evaluate the influence of pH on the reaction product, we performed the same reactions containing an additional 2.0 eq NaNO_3_, except for the pH value. As revealed by the SEM images, left-handed superhelices ***R*-M** can be obtained at pH 3.0–3.8 (Appendix A). At pH 4.0, non-helical fibers were dominant. When the pH was further increased to 4.4, pure right-handed superhelices ***R*-M** were obtained. Clearly, with additional salt of NaNO_3_ in the reaction mixture, the pH value to obtain ***R*-P** is much higher. Therefore, the chiral sense of superhelices can be controlled by both pH and nitrate salt. 

To understand the mechanism of salt-induced chirality inversion, we monitored the hydrothermal reaction products of mixtures containing 2.0 eq NaNO_3_ at pH 3.8 and 120 °C for different periods of time. When the reaction time was less than 30 min, only nanoparticles appeared on the surface of *R*-pempH_2_ ligand (Appendix A). The nanoparticles grew with increasing reaction time. At 3 h, nanorods with a diameter of ca. 150 nm and a length of ca. 500 nm were observed, which have a P/Gd ratio of ca. 2.9/1, very close to that for ***R*-M** (Appendix A). After 4 h of reaction, we obtained a pure phase of left-handed superhelix **R-M**, confirmed by the SEM, EDS, IR, and PXRD measurements (Appendix A). The average width of ***R*-M** increased with prolonging the time, from ca. 0.5 μm at 4 h to 2 μm at 12 h. The results indicate that the left-handed superhelix ***R*-M** is formed from the very beginning. This means that the nitrate salt is involved in the self-assembly process at the nucleation stage. In general, the role of the nitrate anion can be three-fold. The first is its coordination capability towards metal ions. The second is its ability to form an extensive hydrogen bonding network with phosphonate ligands and water molecules. The third is its ability to adjust the ionic strength of the reaction mixture. Since the nitrate anion is uncoordinated in the present cases, the latter two factors should be more important. The existence of extra nitrate anions may facilitate the H-bonding interaction between the nitrate anions and chains, which, in turn, stabilizes the positively charged chains at relatively high pH in order to keep the overall charge balanced. The tendency to maintain NO_3_^−^ anions in the lattice and meanwhile release protons from phosphonate groups at pH 3.8 may cause a mismatch between the chains, leading to the twisted growth of the chains and the formation of ***R*-M** [63]. However, we noticed that ***R*-M** appeared as a minor phase when 1.2 eq NaNO_3_ was added, and as a pure phase when 2.0–3.0 eq NaNO_3_ was added. The concentration of salt is related to the ionic strength of the reaction mixture according to the equation: $I=1/2\sum c_{i}z_{i}^{2}$, where *I* is the ionic strength, *c_i_* is the concentration of various ions, and *z_i_* is the number of the charges carried by each ion. Increasing the ionic strength will increase the Debye−Hückel shielding, reduce the electronic attraction between Gd^3+^ and pempH^−^, and improve contact opportunities between Gd^3+^ and nitrate anion. Therefore, it is the combined effect of the hydrogen bonding ability of the nitrate anion and the ionic strength that is responsible to the salt-induced chirality inversion of the macroscopic superhelices of the present Gd coordination polymers (Figure 3). 

## 3. Conclusions

In summary, we systematically investigated the role of pH and salt in regulating the morphology of the products in a Gd(NO_3_)_3_/*R*-pempH_2_ system. The products of ***R*-Block, *R*-M, *R*-P** and ***R*-Rod** were obtained via pH and salt modulation. This is because pH determines the deprotonation degree of the ligand, and NO_3_^−^ modulates the interchain interaction. Without extra NO_3_^−^, chirality inversion from the left-handed superhelix ***R*-M** to the right-handed superhelix ***R*-P** can be achieved by pH modulation from 3.2 to 3.8. However, ***R*-M** was obtained only as a minor phase. The addition of NaNO_3_ (2.0 eq) at pH 3.8 results in an inversion of chiral sense from ***R*-P** to ***R*-M** as a pure phase. The important role nitrate salt plays in the helicity inversion of the present system may originate from the strong ability of the nitrate anion as a hydrogen bonding receptor and the increased ionic strength of the reaction mixture which improved the nitrate-chain interactions. To our knowledge, this is the first example of salt-assisted macroscopic helical inversion in artificial systems. This work provides a new strategy for the design and synthesis of functional coordination polymers with desired macroscopic helicity. 

## 4. Materials and Methods

### 4.1. Materials and Physical Measurements

R−(1−phenylethylamino)methylphosphonic acid (*R*-pempH_2_) was synthesized according to procedures given in the previous literature [64]. All reagents were purchased from commerical sources without further purification. The morphologies of suprehelices were characterized on a scanning electron microscope (SHIMADZU, SSX-550, Kyoto, Japan) at an acceleration voltage of 5 kV. The UV–Vis spectra of products were recorded on a Perkin Elmer Lambda 950 UV/VIS/NIR spectrometer (Perkin Elmer, Waltham, MA, USA) using powder samples. Elemental analyses were carried out on a PE 240C analyzer (Perkin Elmer, America). Powder X-ray diffraction (PXRD) data were obatined on a Bruker D8 advance diffractometer with Cu-K_α_ radiation. Fourier infrared spectra were recorded on a Bruker Tensor 27 spectrometer (Bruker, Leipzig, Germany) with pressed KBr pellets in the range 4000–400 cm^−1^. Thermogravimetric (TG) analyses were measured on a Mettler Toledo TGA/DSC instrument(Mettler 5MP/PF7548/MET/400W, Mettler-Toledo, Greifensee, Switzerland) in the range of 30–600 °C under a nitrogen flow (20 mL/min) at a heating rate of 5 °C min^−1^. CD spectra of products were recorded on a circular dichroism spectrophotometer (JASCO, J-810, Tokyo, Japan) at room temperature. VCD spectra were recorded on a Bruker VERTEX 80v Fourier transform infrared spectrometer (Bruker, Germany) equipped with a PMA 50 VCD/IRRAS module (Bruker, Germany).

### 4.2. Synthesis of (H_3_O)[Gd_6_(R-pempH_2_)_3_(R-pempH)_15_] (NO_3_)_4_∙9H_2_O (**R*-Block***)

A mixture of *R*-pempH_2_ (0.1080 g, 0.5 mmol) and Gd(NO_3_)_3_·6H_2_O (0.0451 g, 0.1 mmol) was stirred for half an hour in 6 mL H_2_O. Later, the pH was adjusted to 2.9 with 0.5 mol/L NaOH. The glass bottle was then kept in a 15 mL Teflon-lined autoclave, adding 2.0 mL deionized water in the outside of the glass bottle, and heated at 120 °C for 24 h. After cooling to the room temperature, block-shaped colorless crystals of ***R*-Block** were obtained, which were washed with deionized water several times and collected. Yield: 17.3% (15.1 mg) based on Gd(NO_3_)_3_. Elemental analysis (%) calculated for C_162_H_238_N_22_O_66_P_18_Gd_6_·10H_2_O: C, 37.21; H, 4.94; N, 5.89. Found: C, 37.09; H, 4.98; N, 5.90. IR (KBr, cm^−1^): 3446 (w), 3066 (m), 3023 (m), 2986 (m), 2790 (m), 2523 (w), 2405 (m), 1622 (m), 1497 (m), 1456 (m), 1431 (m), 1384 (m), 1313 (m), 1292 (m), 1158 (s), 1083 (s), 1010 (s), 986 (s), 819 (w), 766 (m), 751 (m), 702 (m), 565 (m), 536 (m), 506 (m), 473 (m). Thermal analysis confirmed the removal of ten water molecules below 100 °C (obs. 3.6%, calcd. 3.4%). The number of water molecules is less than that determined by single crystal structural analysis, possibly due to the loss of two lattice water molecules in air. 

### 4.3. Synthesis of Gd[(R-pempH)_2.70_(R-pempH_2_)_0.30_](NO_3_)_0.30_·2H_2_O (**R*-M***)

***R*-M** was obtained following a similar procedure except that pH was adjusted to 3.2. A white powder of ***R*-M** was obtained, which was washed with deionized water several times and collected. Yield: 3.0% (2.5 mg) based on Gd(NO_3_)_3_. Elemental analysis (%) calculated for C_29_H_39.3_N_3_O_9_P_3_Gd·0.30NO_3_·2H_2_O: C, 37.92; H, 5.07; N, 5.42. Found: C, 37.93; H, 5.03; N, 5.58. IR (KBr, cm^−1^): 3426 (w), 3065 (m), 3036 (m), 2987 (m), 2794 (m), 2522 (w), 1623 (m), 1493 (m), 1456 (m), 1384 (m), 1314 (w), 1292 (w), 1273 (w), 1151 (s), 1079 (s), 1017 (s), 985 (s), 939 (m), 822 (w), 765 (m), 751 (m), 702 (m), 565 (m), 536 (m), 504 (m), 472 (m). Thermal analysis confirmed the removal of two water molecules below 120 °C (obs. 3.9%, calcd. 4.2%). 

### 4.4. Synthesis of Gd[(R-pempH)_3_]·2H_2_O (**R*-P***)

***R*-P** was obtained following a similar procedure except that pH was adjusted to 3.8. A white powder of ***R*-P** was obtained, which was washed with deionized water several times and collected. Yield: 18.8% (15.7 mg) based on Gd(NO_3_)_3_. Elemental analysis (%) calculated for C_27_H_39_N_3_O_9_P_3_Gd·2H_2_O: C, 38.80; H, 5.15; N, 5.03. Found: C, 38.78; H, 5.03; N, 5.05. IR (KBr, cm^−1^): 3430 (w), 3060 (m), 3032 (m), 2984 (m), 2787 (w), 2526 (w), 1627 (m), 1497 (m), 1482 (m), 1456 (m), 1384 (m), 1314 (w), 1290 (w), 1277 (w), 1151 (s), 1082 (s), 1018 (s), 986 (s), 941 (m), 818 (w), 763 (m), 752 (m), 701 (m), 564 (m), 535 (m), 503 (m), 470 (m). Thermal analysis confirmed the removal of two water molecules below 120 °C (obs. 4.6%, calcd. 4.3%).

### 4.5. Synthesis of Gd[(R-pempH)_2.70_(R-pempH_2_)_0.30_](NO_3_)_0.30_·2H_2_O (**R*-M’***)

***R*-M’** was obtained following a similar procedure to ***R*-P** except that 2.0 eq NaNO_3_ was added to the reaction mixture. White powder of ***R*-M’** was obtained, washing with deionized water for several times and collecting. Yield: 16.7% (14.3 mg) based on Gd(NO_3_)_3_. Elemental analysis (%) calculated for C_29_H_39.3_N_3_O_9_P_3_Gd·0.30NO_3_·2H_2_O: C, 37.94; H, 5.07; N, 5.41. Found: C, 37.94; H, 4.89; N, 5.66. IR (KBr, cm-1): 3422 (w), 362 (m), 3036 (m), 2983 (m), 2794 (m), 2515 (w), 1623 (m), 1496 (m), 1455 (m), 1384 (m), 1315 (w), 1292 (w), 1274 (w), 1152 (s), 1083 (s), 1019 (s), 985 (s), 923 (w), 822 (w), 765 (m), 751 (m), 699 (m), 566 (m), 536 (m), 505 (m), 473 (m). Thermal analysis confirmed the removal of two water molecules below 120 °C (obs. 4.3%, calcd. 4.2%). 

### 4.6. Single Cryatal X-ray Crystallography 

Single crystal data of ***R*-Block** were collected on a Bruker APEX DUO (for R-Block) diffractometer using graphite-monochromated Mo Kα radiation, λ = 0.71073 Å. The data were integrated using the Siemens SAINT program [65]. Adsorption corrections were applied. The structure was solved by direct methods and refined on *F*^2^ by full-matrix least-squares using SHELXTL [66]. All non-hydrogen atoms were refined anisotropically. All hydrogen atoms bound to carbon were refined isotropically in riding mode. For ***R*-Block**, hydrogen atoms of water molecules were detected via experimental electron density and then refined isotropically with a reasonable restriction of O-H bond distances and H-O-H angles. The residual electron densities were of no chemical significance. The crystallographic data are shown in Appendix A, and the selected bond lengths and angles are shown in Appendix A. Deposition Number 2086976 contains the supplementary crystallographic data. These data can be obtained free of charge from the Cambridge Crystallographic Data Centre via www.ccdc.cam.ac.uk/data_request/cif (accessed on 3 June 2021). 

## Data Availability

Data are available from the authors.

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
