# Peer review of "pH and Salt-Assisted Macroscopic Chirality Inversion of Gadolinium Coordination Polymer"

_molecules, 2022, doi:10.3390/molecules28010163_

Round 1

Reviewer 1 Report

The Authors presented an interesting results of stereocontrol by pH and anion addition. Despite that it is difficult to predict any practical application of collected knowledge it might be taking into the consideration while design new materials for the optical and liquid crystal applications.

The manuscript is carefuly prepared and only minor concerns could be made:

1) The Hirshfeld surface for the crystal structure and 2D fingerprint plot analysis could be added to the data set.

2) the atom contacts in the basic units (for  P, M and rod forms) of the chain could be presented in a separate image and discussed including all the atoms of the ligands.

Tking in to the consideration good quality of data I would recommend publication after the minor revision.

Reviewer 2 Report

The manuscript reports preparing a Gd-coordination polymer system that shows chirality inversion by pH and the addition of NO3. All compounds are characterized, and the unique phenomena will attract many readers in this area. I want to request that to add comments on the other anions. I recommend that this manuscript be accepted in this journal.
